# A bilateral filtering-based image enhancement for Alzheimer disease classification using CNN

**Nicodemus Songose Awarayi** [1,2]*, **Frimpong Twum**[1], **James Ben Hayfron-Acquah**[1], **Kwabena Owusu-Agyemang**[1]

**1** Department of Computer Science, Kwame Nkrumah University of Science and Technology, Kumasi, Ghana, **2** Department of Computer Science and Informatics, University of Energy and Natural Resources, Sunyani, Ghana

* nicodemus.awarayi@uenr.edu.gh

**Data Availability Statement:** Data cannot be shared publicly because of institutional data access policy. Data are available from the ADNI Institutional Data Access / Ethics Committee

## Abstract

This study aims to develop an optimally performing convolutional neural network to classify Alzheimer's disease into mild cognitive impairment, normal controls, or Alzheimer's disease classes using a magnetic resonance imaging dataset. To achieve this, we focused the study on addressing the challenge of image noise, which impacts the performance of deep learning models. The study introduced a scheme for enhancing images to improve the quality of the datasets. Specifically, an image enhancement algorithm based on histogram equalization and bilateral filtering techniques was deployed to reduce noise and enhance the quality of the images. Subsequently, a convolutional neural network model comprising four convolutional layers and two hidden layers was devised for classifying Alzheimer's disease into three (3) distinct categories, namely mild cognitive impairment, Alzheimer's disease, and normal controls. The model was trained and evaluated using a 10-fold cross-validation sampling approach with a learning rate of 0.001 and 200 training epochs at each instance. The proposed model yielded notable results, such as an accuracy of 93.45% and an area under the curve value of 0.99 when trained on the three classes. The model further showed superior results on binary classification compared with existing methods. The model recorded 94.39%, 94.92%, and 95.62% accuracies for Alzheimer's disease versus normal controls, Alzheimer's disease versus mild cognitive impairment, and mild cognitive impairment versus normal controls classes, respectively.

## Introduction

Alzheimer's disease (AD) is a neurological condition contributing to the progressive degeneration of cognitive abilities, including memory, visual-spatial perception, and alterations in personality and behavior. This ailment is commonly linked to the geriatric population, although it can impact individuals of all ages. It is postulated that a global population exceeding 1.5 million individuals exhibits indications of this particular ailment, with a probable escalation anticipated within the forthcoming two to three decades [1]. The absence of a viable remedy for AD underscores the importance of timely identification to manage the condition and forestall

(contact via adni.loni.usc.edu) for researchers who meet the criteria for access to confidential data.

**Funding:** The author(s) received no specific funding for this work.

**Competing interests:** The authors have declared that no competing interests exist.

further deterioration effectively. Individuals in the initial stages of the disease, referred to as mild cognitive impairment (MCI), primarily manifest indications such as memory impairment or diminished decision-making abilities, which may subsequently deteriorate into AD. Detecting MCI early enough would be beneficial for clinicians in preventing further progression [2, 3].

Brain imaging is a known non-invasive technique for detecting diseases through the use of brain scans in the form of magnetic resonance imaging (MRI), computed tomography (CT), or positron emission tomography (PET) [4, 5]. Brain imaging has substantially contributed to the timely diagnosis of AD and detecting the transformation of MCI into AD using a deep learning methodology. Various alternative deep-learning methods have been successfully implemented to accurately categorize AD, including those of Y. Zhang et al. [6], Zhang et al. [7].

Despite these notable achievements, using MRI imaging to diagnose AD or predict its conversion using deep learning is still challenging. One of the key obstacles is the insufficiency of appropriate and uncontaminated data, which adversely impacts the training of deep learning models and consequently undermines their accuracy and generalizability [8–12]. As the data is collected over time, there is a tendency for the datasets not to be enough for training models, and again, due to the varying image scanning protocols, imaging artefacts, and moving artefacts being used during the data collection, MRI images are highly susceptible to noise. Image quality can also vary due to the different imaging equipment being used. To address these challenges, it requires developing more robust deep learning models that are more tolerant to noisy images [13, 14] or applying a data preprocessing approach to addressing image quality [15–18].

The motivation of our study is to propose a more reliable approach to improving image quality with ease for classifying AD with improved accuracy. As such, the research devised a proficient strategy for improving image quality by utilizing bilateral filtering and histogram equalization techniques. The proposed method aimed to reduce noise and enhance brightness in images, thereby improving the quality of images used in categorizing AD. The primary contribution of this study comprises two key components:

- First, an image enhancement technique that effectively reduces noise in MRI images while preserving image quality was designed and implemented. The image enhancement algorithm for color images was developed using two image processing techniques: image equalization and bilateral filtering. The algorithm mainly improves the quality of the images while maintaining their edges and details. The bilateral filtering was used to keep the edges and details of the image, while histogram equalization modifies the intensities of the image to improve its contrast.

- Second, a CNN model consisting of four convolutional layers and two hidden layers successfully classifies AD more accurately than existing deep learning models. The model was trained using the k-fold cross-validation technique, ensuring that all the images were effectively used in training and testing the model.

The remaining sections of the paper are the literature review, materials and methods, results, and discussion. The literature review section presents a review of some existing state-of-the-art studies. A description of the dataset, the proposed method, and a detailed description of the experiment are presented under the materials and methods section. The results section reveals the study's experimental results, which are further discussed in the discussion section.

## Literature review

This section analyzes prior studies that employed MRI images to construct machine learning and deep learning models to classify AD. A particular emphasis was placed on methods that address data inadequacy and image quality challenges.

Ebrahimi et al. [19] designed a pre-trained ResNet in their deep sequencing model to classify AD. The prognostication of the AD recorded an accuracy of approximately 91.78%, signifying a 10% enhancement compared to other CNNs. Kang et al. [20] developed a deep ensemble 2D CNN architecture incorporating multiple models and slides. The model's accuracy was 90.36% on normal controls (NC)/AD, 77.19% on AD/MCI classes, and 72.36% on NC/MCI class labels.

Alinsaif and Lang [21] designed a methodology that integrated 3D shearlet-based descriptors to decrease the dimensionality of MRI datasets. The method showed a noteworthy degree of efficacy in identifying AD. Mahendran and P. M. [22] developed a framework to detect AD that achieved an accuracy of 88.7% using a limited number of data records.

Zhang et al. [23] proposed a multi-modal model to facilitate timely AD detection. The proposed method showed superior performance in diagnosing AD, as evidenced by its training on two image modalities, namely PET and MRI, obtained from ADNI.

Spasov et al. [24] deployed a parameter-efficient model to check the conversion of MCI to AD. The approach combines various data types, including MRI images, demographic information, neuropsychological data, and genetic data. The method showed adaptability and has the potential to incorporate additional imaging techniques, including PET images and various collections of medical information.

In their scholarly work, Sharma et al. [25] presented a model capable of extracting all-level features. The model, FDN-ADNet, designed for early AD diagnosis using MRI images, yielded encouraging outcomes. Abrol et al. [26] applied ResNet in analyzing medical images to investigate the conversion of MCI to AD. Initially, the deep models were trained solely on MCI individuals for prediction purposes. Subsequently, a domain transfer learning version was employed, which involved additional training on AD and NC. The frameworks demonstrated significant prominence in the use of the model. The authors of [27] tackled the issue of insufficient data and missing data in diagnosing AD. The accuracy of all four classes of subjects used was observed to have improved by up to 3% when both complete and incomplete samples were considered during the model generation and testing phases.

In their study, Liu et. al. [28] presented a CNN-based approach for detecting AD. The model was specifically designed to train on a limited quantity of MRIs and accurately identify cases of AD. The model's accuracy was approximately 78.02%, and it was observed that its portability was enhanced as an added advantage. Janghel and Rathore [29] applied a distinctive approach to improve the efficiency of Convolutional Neural Network (CNN) models. They implemented a preprocessing technique on an image dataset before submitting it to the CNN architecture for feature extraction. The experimental findings indicate that an fMRI dataset yielded a precision of 99.95%, whereas the PET data stood at 73.46%.

In their study, Bae et. al. [30] employed transfer learning to forecast the progression of MCI to AD dementia using a 3-D CNN. An 82.4% accuracy recorded for the target task surpassed the performance of existing models in the respective domain. Sathiyamoorthi et al. [31] proposed a method that exploits an adaptive histogram adjustment image correction algorithm to improve the quality of the image in terms of brightness and contrast. The algorithm was applied to segment the interested AD regions in an adaptive and modified manner. The diverse characteristics were computed using a second-order Gray Level Co-Occurrence Matrix. The method classified diseased images and their respective stages based on distinctive

features. The empirical findings demonstrate that the suggested approach yields superior levels of precision and efficacy compared to the current system.

Ferri et al. [32] proposed a classification system with LORETA source estimation derived from rsEEG and sMRI variables. The study implemented two artificial neural networks (ANNs) to classify participants as having AD or being part of the control group. The results indicated classification accuracy of 83%, and 86% for the two ANNs, respectively.

In their study, Tong et al. [33] developed a CNN-Sparse Coding Network (CNN-SCN) architecture to detect MCI before it converts. The empirical findings indicate that the model exhibits good stability, accuracy, and generalization. The experiment recorded an accuracy rate of 92.6% in the AD/CN class, while the AD/MCI and MCI/CN classes recorded an accuracy of 74.9% and 76.3%, respectively.

Hridhee et al. [34] investigated and proposed an early detection 2D CNN model for classifying AD using MRI data samples. The researchers applied various image processing and augmentation strategies to the dataset. Subsequently, they trained the data on a VGG16, Xception, and a custom model. The customized model achieved the best performance of the selected models with an accuracy of 94.77%.

Allada et al. [35] aimed to enhance the accuracy of classifying AD and developed a swarm multi-verse algorithm to optimize a deep neuro-fuzzy network for AD classification at various stages. Their study incorporated techniques such as median filtering for preprocessing to improve image quality, a channel-wise feature pyramid network for image segmentation, and CNN for feature extraction. Their model achieved an accuracy of 89.9%, assessed based on k-fold values.

Gowhar et al. [36] introduced a deep learning framework centered on CNN for early AD detection. They implemented various data preprocessing and augmentation techniques on the dataset. The classifier was constructed through feature extraction, reduction, and classification. Their proposed model surpassed existing models, achieving an accuracy of 96.22%.

Tufail et al. [37] developed an early-stage AD classifier using 2D and 3D CNN with PET images. Their study differentiated between binary and multiclass categories of AD. Experimental results highlighted superior performance with an accuracy of 89.21% for the AD/NC binary classification using the 3D-CNN model. They employed data augmentation and 5-fold cross-validation to boost the model's performance.

This section examined the performance of some deep learning models in classifying AD and is summarized in Table 1.

## Materials and methods

### Subjects/datasets

The datasets were acquired from the AD Neuroimaging Initiative (ADNI) database (adni.loni.usc.edu) [38]. In 2003, the ADNI was established as a public-private collaboration with the primary objective of evaluating the feasibility of integrating MRI, PET, and other biological markers to investigate and curb the degeneration of MCI to AD.

This study extracted MRI images from a sample of 396 subjects diagnosed with AD, 255 subjects classified as NC, and 104 subjects with MCI from the ADNI database captured from 2005 to 2021 and within an age range of 55 to 97 years old. The MRI datasets were initially in DICOM format and were subsequently transformed into JPG format. The images collected for the experiment included 1,581 AD images, 1,310 MCI images, and 1,591 NC images. Each image was initially dimensioned as 256 x 256 pixels and later resized to 64 x 64 pixels.

**Table 1. Summary of existing CNN models.**

| S/No. | Author | Cite | Method | Dataset | Classification | Accuracy |
|---|---|---|---|---|---|---|
| 1 | (Hridhee et al., 2023) | [34] | CNN | ADNI | AD/NC/MCI/ EMCI/LMCI | 94.77 |
| 2 | (Allada et al., 2023) | [35] | Neuro-fuzzy | Kaggle | AD/NC/MCI/ EMCI/LMCI | 89.9 |
| 3 | (Gowhar et al., 2023) | [36] | CNN | ADNI | AD/NC/MCI/ EMCI/LMCI | 96.22 |
| 4 | (Tufail et al., 2022) | [37] | CNN | ADNI | AD/NC | 89.21 |
| | | | | | AD/MCI | 71.70 |
| | | | | | NC/MCI | 62.25 |
| | | | | | AD/NC/MCI | 59.73 |
| 5 | (Kang et al., 2021) | [20] | CNN | ADNI | AD/NC | 90.36 |
| | | | | | AD/MCI | 77.19 |
| | | | | | NC/MCI | 72.36 |
| 6 | (Mahendran & P, 2022) | [22] | EDRNN | GEO Omnibus | AD/MCI | 88.7 |
| 7 | (Tong et al., 2022) | [33] | CNN | ADNI | AD/NC | 92.6 |
| | | | | | AD/MCI | 74.9 |
| | | | | | NC/MCI | 76.3 |
| 8 | (Zhang et al., 2022) | [6] | ResNet | ADNI | AD/NC | 90 |
| | | | | | AD/MCI | 74.9 |
| | | | | | NC/MCI | 62.6 |
| 9 | (J. Liu et al., 2021) | [28] | CNN | OASIS | AD/NC/ MCI | 78.02 |
| 10 | (Ferri et al., 2021) | [32] | autoencoders | ADNI | AD/NC | 89.0 |

## Method

In this study, we sought to build a deep learning model with optimal performance to classify AD into three categories: MCI, AD, and NC. To achieve this, we proposed an approach to improve the quality of the image datasets during the data preprocessing stage, after which the data volume was increased using data augmentation. The data was then sampled using the $K$-fold cross-validation technique to ensure that all data samples were used in the training and testing. The method divides the datasets into $K$ folds, of which one-fold is used for testing and the remaining for training. The technique assesses the model's capability to handle new examples, with $K$ referring to the number of groups into which the data samples are divided. As presented in the subsequent section, a convolutional neural network was proposed, which was trained and evaluated iteratively using 10-fold cross-validation. For each iteration, the model is tested with the one-fold test data, after which the recorded test results are averaged. A detailed representation of the flow of activities in the proposed method is shown in Fig 1.

The experiment was implemented with an NVIDIA GeForce GTX 1060 Graphic Processing Unit (GPU) machine operating on CUDA 10.1 with 8 GB of memory allocation. The model was implemented in Python using the TensorFlow framework and other libraries such as Sklearn, Numpy, Keras, Pillow and Opencv-python.

## Data preprocessing

Data preprocessing is an essential step in conducting a deep learning experiment, as the efficacy of the dataset can significantly influence the learning outcomes of the models. The MRI image datasets utilized in this study were initially transformed into JPG format to facilitate manual preprocessing of the datasets. MRI images that were corrupt or of insufficient quality were manually removed. Following manual preprocessing, it was observed that the dataset still

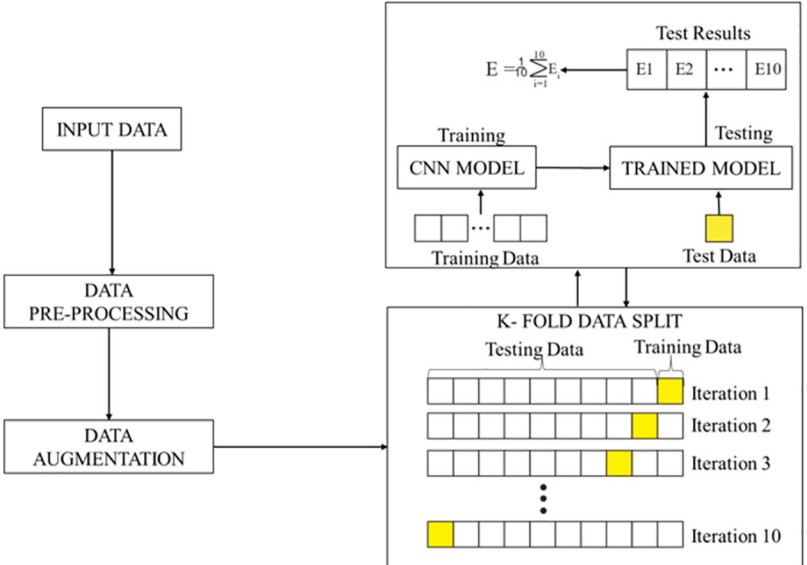

**Fig 1. The proposed method defining the flow of activities.**

contained noisy images with varying brightness levels, highlighting the necessity for additional preprocessing measures.

An algorithm was proposed to address the issue of image noise and varying brightness. The algorithm was designed with histogram equalization and bilateral filtering, as presented under Algorithm 1.

**Algorithm 1** Image enhancement algorithm

```
I ← RGBimage
[I_r, I_g, I_b] ← GetImageChannels(I)
I_r ← histogramEqualize(I_r)
I_g ← histogramEqualize(I_g)
I_b ← histogramEqualize(I_b)
I_r ← bilateralFilter(I_r)
I_g ← bilateralFilter(I_g)
I_b ← bilateralFilter(I_b)
I ← mergeImageChannels(I_r, I_g, I_b)
I ← normalizedImage(I)
```

Algorithm 1 was designed to improve the contrast of an input image by performing histogram equalization on each color channel, followed by bilateral filtering on the equalized channels. The bilateral filtering aims to maintain the edges and details of the image. The filtered channels are recombined into a unified RGB image and subjected to normalization procedures to enhance overall contrast. Eq (1) defines bilateral filtering.

$$\text{BF}[\text{I}]_p = \frac{1}{W_p} \sum_{q \epsilon s} G_{\sigma_s}(||p-q||) G_{\sigma_r}(|I_p - I_q|) I_q \tag{1}$$

$$\text{W}_p = \sum_{q \epsilon s} G_{\sigma_s}(||p-q||) G_{\sigma_r}(|I_p - I_q|) I_q \tag{2}$$

The equation for the image value at pixel position $p$ is defined as $I_p$, where $G_\sigma$ represents a 2D Gaussian kernel. The normalization factor, $W_p$, is utilized to ensure that the pixel weights sum

up to 1.0. The parameters *s* and *r* determine the extent of filtering applied to the image. Eq (3) defines the histogram equalization process, which is used to modify an image's intensities to improve its contrast.

$$I_i, j = floor(L-1) \sum_{n=0}^{f_i j} P_n \qquad (3)$$

$$P_n = \frac{I_n}{T_p} \quad n = 0, 1, \dots, L-1 \qquad (4)$$

where $L$ is the number of possible intensity values, $I_n$ is the number of pixels with intensity, $n$ and $T_p$ is the total number of pixels.

## Convolutional neural network model

The CNN model in Fig 2 comprised four convolutional layers to extract features from 64 x 64-sized color image inputs. In this model, every convolutional layer is paired with a corresponding max-pooling layer, batch normalization layer, and a dropout layer to reduce the learnable parameters. Each layer also incorporated a rectified linear activation function, a 3 by 3 kernel filter and L2-regularization. All convolutional layers had a stride of one (1), and same padding was used to keep the dimensions of the output feature map the same as the input

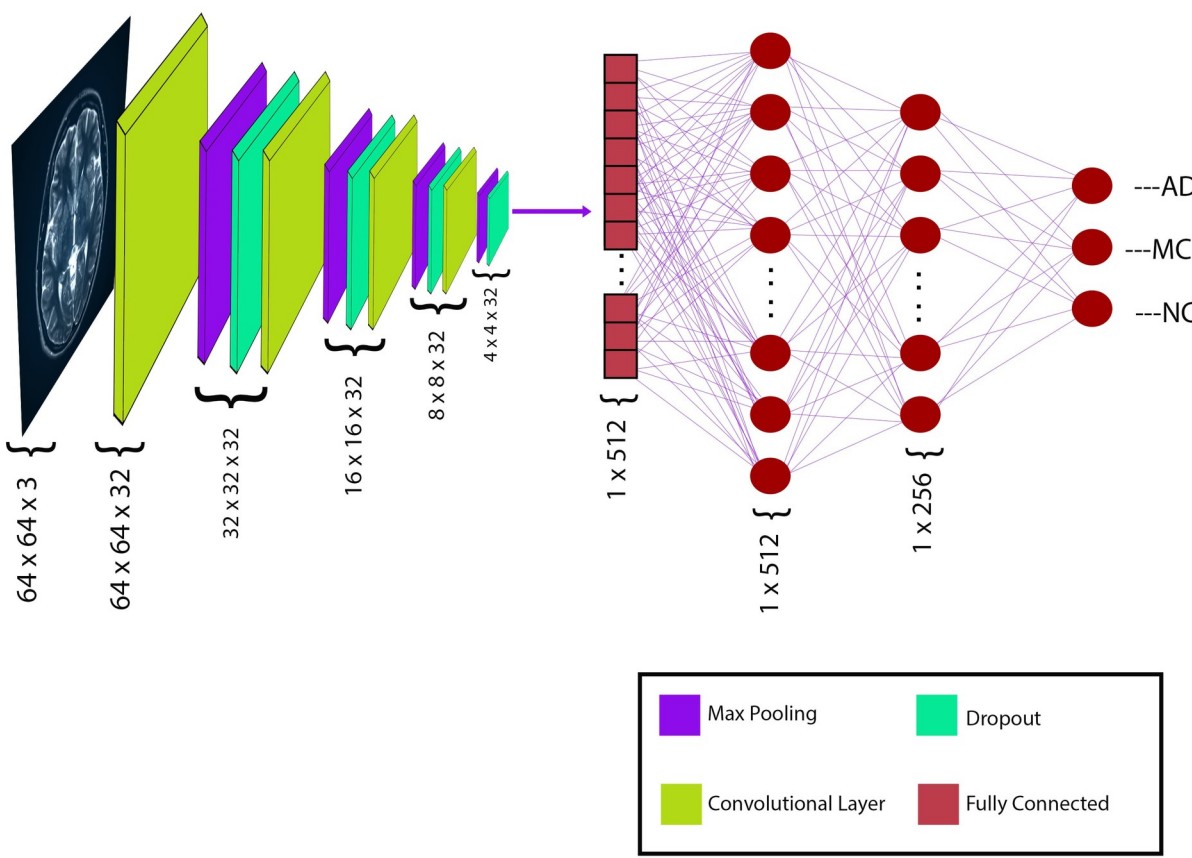

**Fig 2. The proposed CNN model for classifying Alzheimer's disease.**

dimension. Two (2) hidden layers were incorporated into the model, wherein a batch normalization layer and a dropout layer succeeded every layer. The model generated 426,979 parameters, consisting of 425,187 trainable and 1,792 nontrainable. Additional hyperparameters include a learning rate of 0.001, a batch size of 64, and 200 training epochs.

## Performance evaluation

Various metrics were used to measure the model's performance during training and testing. These include recall, precision, accuracy and area under the curve (AUC).

The accuracy metric represents the aggregate count of accurately predicted labels by the model and is defined as Eq (5).

$$\text{accuracy} = \frac{TP + TN}{TP + TN + FP + FN} \tag{5}$$

TN is the total number of true positives, TN is the true negatives, FP refers to the false positives, and FN is the false negatives. As shown in Eq (6), precision is the ratio of true positives to the total number of predictions made.

$$\text{precision} = \frac{TP}{TP + TN} \tag{6}$$

Recall calculates the true positive rates as presented in Eq (7).

$$\text{recall} = \frac{TP}{TP + FN} \tag{7}$$

The ROC curve's integral, AUC, corresponds to the likelihood of a positive sample being ranked at random higher than a negative sample. Given that the ROC curve is constructed through the successive joining of coordinates $(x_1, y_1), (x_2, y_2), \ldots, (x_m, y_m)$, the AUC is approximated using Eq (8).

$$\text{AUC} = \frac{1}{2} \sum_{i=1}^{m-1} (x_{i-1} - x_i)(y_i - y_{i+1}) \tag{8}$$

## Results

In this study, an algorithm for enhancing images was developed by integrating two image processing techniques: image equalization and bilateral filtering. The algorithm primarily enhances the image quality while preserving its edges and details. Bilateral filtering is employed to maintain the edges and details of the image, whereas histogram equalization adjusts the image's intensity levels to improve contrast. Fig 3a shows the original images, while Fig 3b shows the images processed by the proposed algorithm.

In order to ascertain the impact of the image transformation by the algorithm, image quality parameters such as mean pixel intensity, contrast, and entropy were used. The mean pixel intensity calculates the average of the pixel intensities across the image, and contrast evaluates the variability (standard deviation) of pixel intensities, contributing to the sharpness or clarity of the image. An increase in mean pixel intensity indicates that the image has become brighter, while a higher contrast suggests enhanced sharpness. Entropy measures the richness of information in the image's intensity distribution, with an increase in entropy indicating an improvement in image quality. The mean pixel intensity, contrast, and entropy values were calculated for both the original and enhanced images and visualized in Fig 4. The results

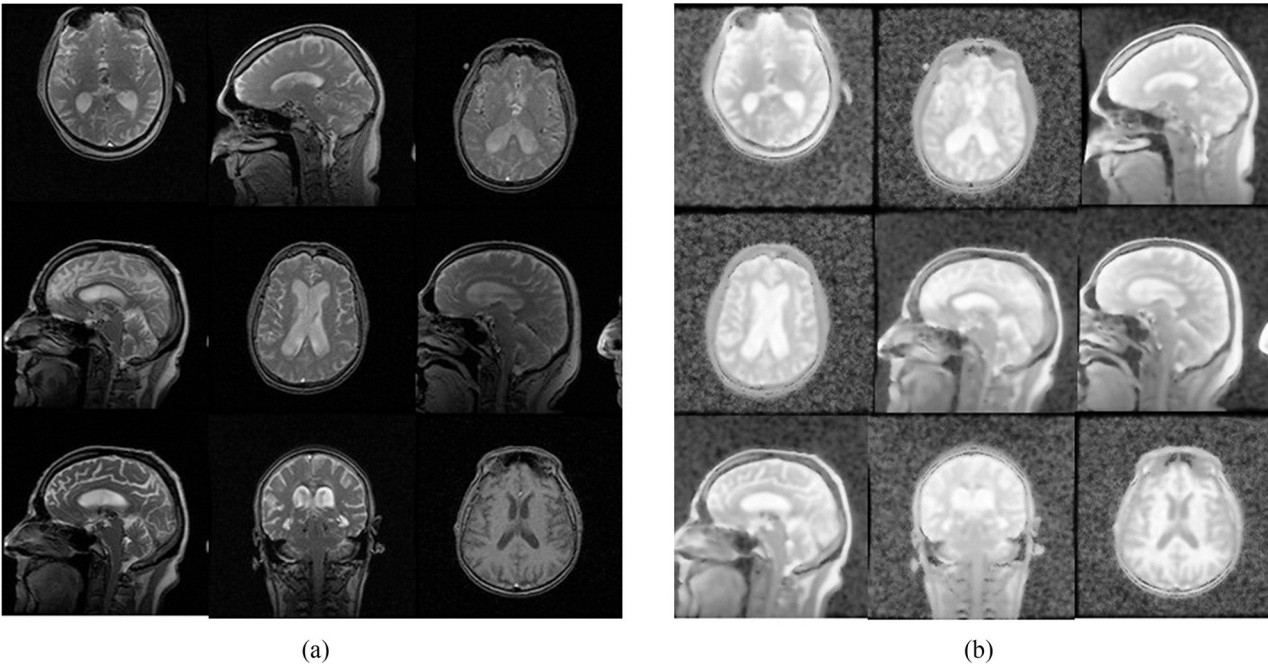

(a)                                        (b)

**Fig 3. Applying the proposed image enhancement algorithm.** (a) Original images (b) Transformed images.

indicate that the enhanced images showed more brightness, sharpness and richness of information since the mean pixel intensity, contrast, and entropy are higher.

This study developed a CNN model to classify AD based on three distinct class labels: AD, MCI, and NC. The study further investigated the binary classification of AD versus MCI, AD versus NC, and MCI versus NC. The model was trained using a 10-fold cross-validation technique, with the learning rate hyperparameter set to 0.001, a batch size of 64, and 200 epochs. The training and testing were done on both the original and enhanced datasets to aid in assessing the impact of the image transformation. The test outcomes are displayed in Table 2.

The model attained an accuracy of 90.34% on the original dataset, while on the transformed dataset, an accuracy of approximately 93.45% was recorded, accompanied by a standard deviation of 1.20%, suggesting the model's capacity to forecast the labels accurately. The precision of the model in identifying true positives was 93.70%, with a standard deviation of 1.21%. The recorded recall value is 93.24%, with a standard deviation of 1.25%. The model achieved a high

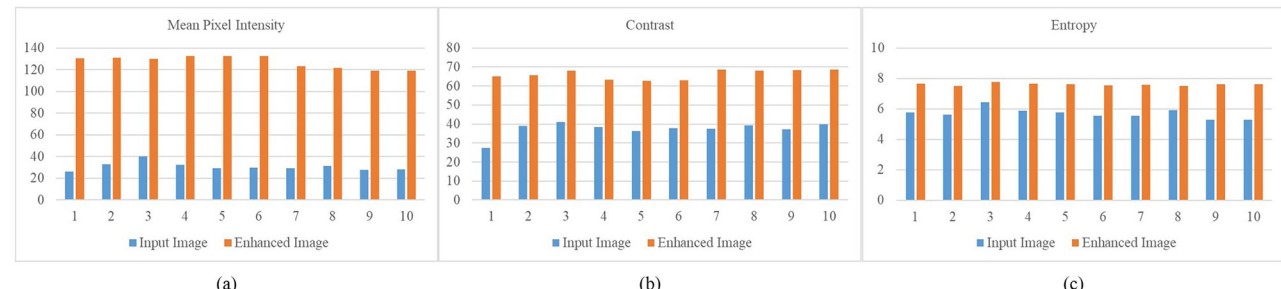

(a)                                        (b)                                        (c)

**Fig 4. Image quality assessment.** (a) Mean pixel intensity (b) Contrast (c) Entropy.

**Table 2. Testing results for classifying AD, MCI and NC.**

| Fold | No Image Enhancement | With Image Enhancement | | | |
|---|---|---|---|---|---|
| | Accuracy (%) | Accuracy (%) | Precision(%) | Recall (%) | AUC |
| 1 | 92.37 | 93.09 | 93.49 | 92.87 | 0.98 |
| 2 | 87.66 | 93.42 | 93.51 | 93.20 | 0.99 |
| 3 | 91.98 | 90.30 | 90.56 | 89.86 | 0.97 |
| 4 | 89.57 | 93.76 | 94.27 | 93.53 | 0.99 |
| 5 | 89.44 | 94.20 | 94.51 | 94.08 | 0.99 |
| 6 | 89.95 | 93.64 | 93.84 | 93.53 | 0.99 |
| 7 | 90.59 | 94.64 | 94.62 | 94.20 | 0.99 |
| 8 | 92.37 | 94.53 | 94.95 | 94.42 | 0.99 |
| 9 | 91.46 | 92.75 | 92.85 | 92.75 | 0.98 |
| 10 | 88.03 | 94.20 | 94.39 | 93.97 | 0.99 |
| Average | 90.34 | 93.45 | 93.70 | 93.24 | 0.99 |
| Standard Deviation | 1.62 | 1.20 | 1.21 | 1.25 | 0.005 |

AUC score of 0.99 and a low standard deviation of 0.005, suggesting the accuracy of the model's predictions. The performance of the model is further visualized using the receiver operating characteristic (ROC) curve and confusion matrix in Fig 5.

The model underwent further training with binary class examples, namely AD and MCI, AD and NC, and MCI and NC, while maintaining the same hyperparameters and model architecture. The model demonstrated significant efficacy in forecasting the diverse labels, as outlined in Table 3. The test results show a consistent accuracy, precision, and recall score. The model's accuracy rates for classifying AD versus MCI, AD versus NC, and MCI versus NC stood at 94.92%, 94.39%, and 95.62%, respectively. Our proposed model demonstrated good performance, as evidenced by AUC scores of 0.98, 0.98, and 0.99 for

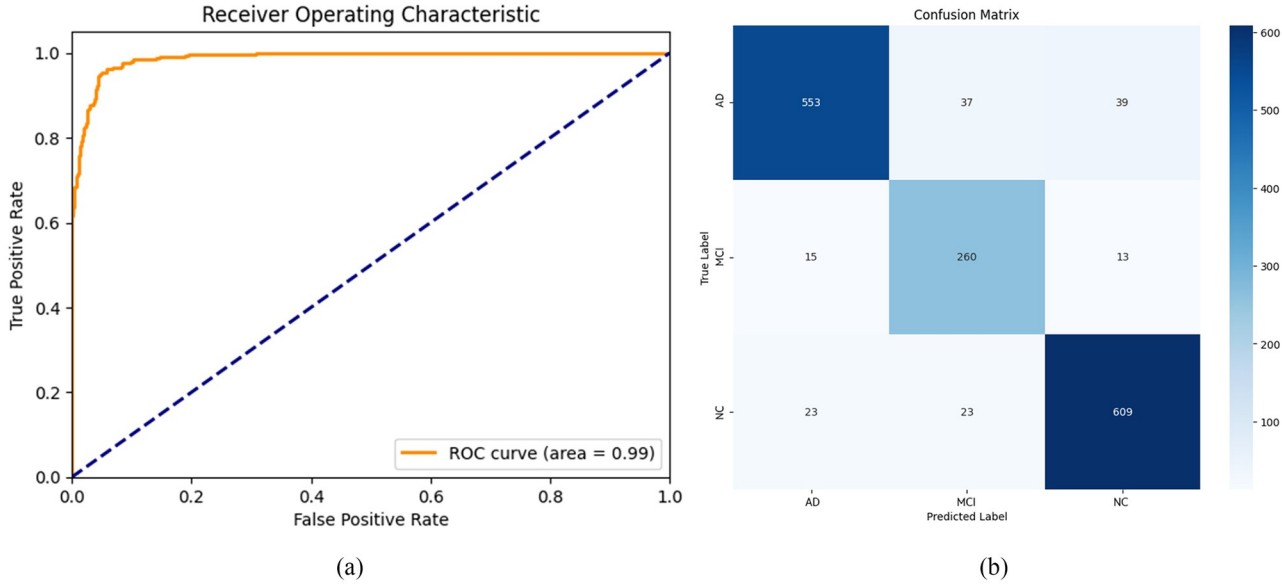

(a)　　　　　　　　　　　　　　(b)

**Fig 5. ROC curve and confusion matrix for classifying AD, MCI and NC.** (a) ROC curve (b) Confusion matrix.

**Table 3. Testing results for classifying AD/MCI, AD/NC and MCI/NC.**

| Fold | AD/MCI | | | | AD/NC | | | | MCI/NC | | | |
|---|---|---|---|---|---|---|---|---|---|---|---|---|
| | Accuracy | Precision | Recall | AUC | Accuracy | Precision | Recall | AUC | Accuracy | Precision | Recall | AUC |
| 1 | 94.30 | 94.30 | 94.30 | 0.97 | 94.02 | 94.02 | 94.02 | 0.99 | 96.04 | 96.04 | 96.04 | 0.99 |
| 2 | 88.77 | 88.77 | 88.77 | 0.94 | 95.28 | 95.28 | 95.28 | 0.98 | 95.01 | 95.01 | 95.01 | 0.99 |
| 3 | 96.19 | 96.19 | 96.19 | 0.99 | 96.54 | 96.54 | 96.54 | 0.99 | 95.86 | 95.86 | 95.86 | 0.99 |
| 4 | 94.64 | 94.64 | 94.64 | 0.98 | 92.91 | 92.91 | 92.91 | 0.98 | 96.38 | 96.38 | 96.38 | 0.99 |
| 5 | 95.16 | 95.16 | 95.16 | 0.99 | 94.79 | 94.79 | 94.79 | 0.98 | 95.52 | 95.52 | 95.52 | 0.99 |
| 6 | 95.85 | 95.85 | 95.85 | 0.99 | 95.90 | 95.90 | 95.90 | 0.98 | 96.03 | 96.03 | 96.03 | 0.99 |
| 7 | 95.67 | 95.67 | 95.67 | 0.99 | 94.01 | 94.01 | 94.01 | 0.98 | 96.38 | 96.38 | 96.38 | 0.98 |
| 8 | 95.85 | 95.85 | 95.85 | 0.98 | 92.27 | 92.27 | 92.27 | 0.98 | 96.21 | 96.21 | 96.21 | 0.98 |
| 9 | 97.75 | 97.75 | 97.75 | 0.99 | 92.43 | 92.43 | 92.43 | 0.98 | 94.31 | 94.31 | 94.31 | 0.98 |
| 10 | 94.98 | 94.98 | 94.98 | 0.98 | 95.74 | 95.74 | 95.74 | 0.99 | 94.48 | 94.48 | 94.48 | 0.99 |
| AVG | 94.92 | 94.92 | 94.92 | 0.98 | 94.39 | 94.39 | 94.39 | 0.98 | 95.62 | 95.62 | 95.62 | 0.99 |
| S. D. | 2.24 | 2.24 | 2.24 | 0.01 | 1.43 | 1.43 | 1.43 | 0.01 | 0.73 | 0.73 | 0.73 | 0.004 |

comparing AD versus MCI, AD versus NC, and MCI versus NC, respectively. The confusion matrix and the ROC curve for the binary classification are presented in Figs 6 and 7, respectively.

To adequately examine the model's performance, we conducted a comparative study with prevailing models employed for AD classification. The accuracy metric was the baseline metric for comparing this study to the existing models. The findings suggest the superior efficacy of our research in AD classification, as demonstrated in Table 4.

## Discussion

The experimental results highlight the superior performance of the proposed method in relation to other methods discussed in the literature, especially in the binary and multiclass classification of AD. As evident from Table 4, the accuracy of the proposed model surpassed that of [6, 20, 22, 32, 33, 37] in binary classification. In multiclass scenarios, our model outperformed the methods from [28, 37] that considered the same number of classes as in this study. Conversely, methods [34, 36] which encompassed more than the three classes in our study

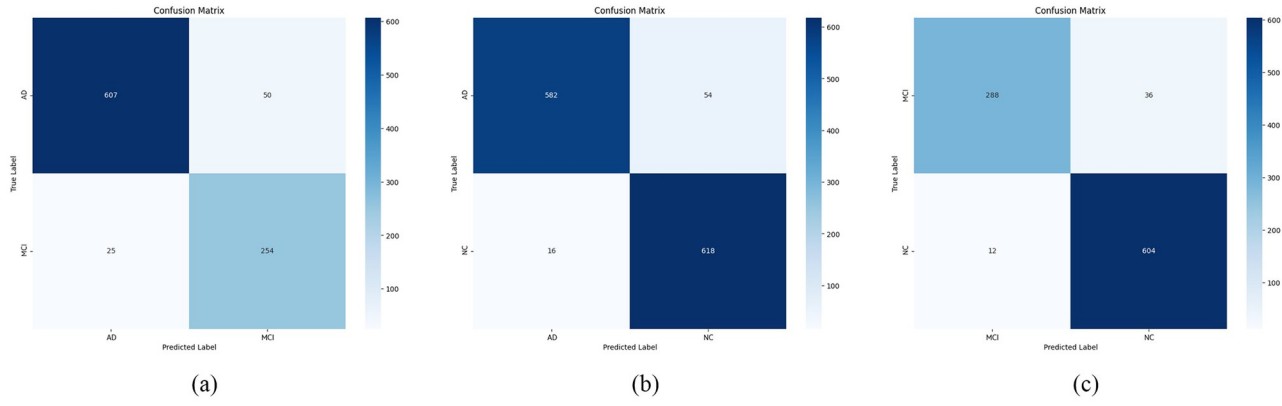

**Fig 6. Confusion matrix for binary classification.** (a) AD/MCI (b) AD/NC (c) MCI/NC.

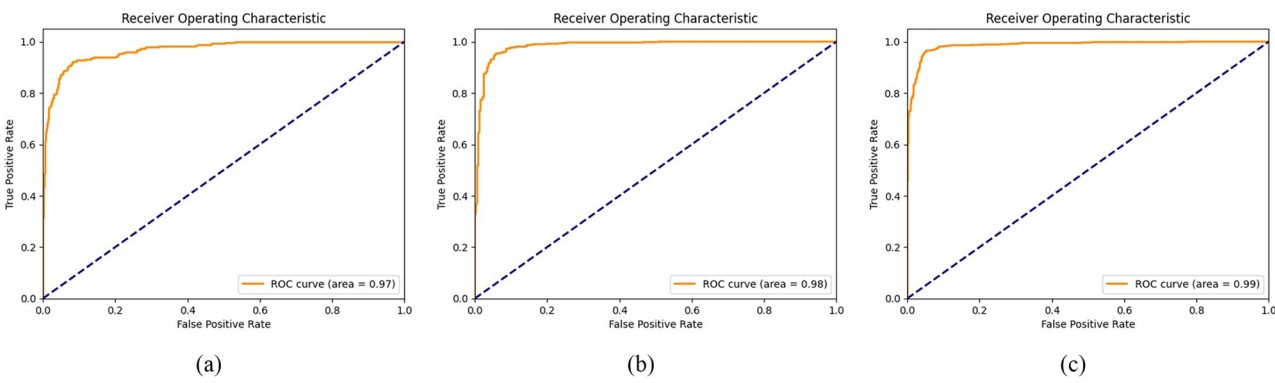

**Fig 7. ROC curve for binary classification.** (a) AD/MCI (b) AD/NC (c) MCI/NC.

exhibited higher accuracies than our proposed model. Thus, the findings suggest that our model stands out as the most effective for binary classification, as well as for multiclass classification, when specifically focusing on AD, MCI, and NC labels.

## Conclusion

AD is a prevalent neurological disorder, predominantly affecting the elderly, and remains without a definitive cure. Early detection of AD is paramount to manage and potentially slow its progression from a mild stage to more severe stages. While deep learning offers promising avenues for detecting AD, it also presents specific challenges, with image quality being a prominent one. The study proposed and implemented an image quality scheme using histogram equalization and bilateral filtering to improve the dataset's quality, which was then used to train the proposed convolutional neural network. The proposed model consisted of four convolutional layers and two fully connected layers. It was trained using 10-fold cross-validation on three class labels: AD, MCI, and NC. The resulting accuracy, precision, recall score, and AUC score stood at 93.45%, 93.70%, 93.24%, and 0.99, respectively. These scores were comparatively higher than those of some selected state-of-the-art existing models.

**Table 4. Comparative analysis of existing and proposed model.**

| S/No. | Various Models | Cite | Class labels | | | |
|---|---|---|---|---|---|---|
| | | | AD/MCI | AD/NC | MCI/NC | Multiclass (AD/MCI/NC or more) |
| 1 | (Hridhee et al., 2023) | [34] | - | - | - | 94.77 |
| 2 | (Allada et al., 2023) | [35] | - | - | - | 89.9 |
| 3 | (Gowhar et al., 2023) | [36] | - | - | - | 96.22 |
| 4 | (Tufail et al., 2022) | [37] | 71.70 | 89.21 | 62.25 | 59.73 |
| 5 | (Kang et al., 2021) | [20] | 77.19 | 90.36 | 72.36 | - |
| 6 | (Mahendran and P M, 2022) | [22] | 88.7 | - | - | - |
| 7 | (Tong et al., 2022) | [33] | 74.9 | 92.6 | 76.3 | - |
| 8 | (Zhang et al., 2022) | [6] | 82.5 | 90.00 | 62.6 | - |
| 9 | (J. Liu et al., 2021) | [28] | - | - | - | 78.02 |
| 10 | (Ferri et al., 2021) | [32] | 89.0 | - | - | - |
| 11 | Proposed Model | | 94.92 | 94.39 | 95.62 | 93.45 |

## Supporting information

**S1 Checklist.** *PLOS ONE* **clinical studies checklist.**
(PDF)

## Acknowledgments

We acknowledge the ADNI for granting us access to use their datasets for this research.

## Author Contributions

**Conceptualization:** Nicodemus Songose Awarayi, Frimpong Twum, James Ben Hayfron-Acquah, Kwabena Owusu-Agyemang.

**Formal analysis:** Nicodemus Songose Awarayi.

**Methodology:** Nicodemus Songose Awarayi.

**Supervision:** Frimpong Twum, James Ben Hayfron-Acquah, Kwabena Owusu-Agyemang.

**Writing – original draft:** Nicodemus Songose Awarayi.

**Writing – review & editing:** Frimpong Twum, James Ben Hayfron-Acquah, Kwabena Owusu-Agyemang.

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
