## [Decision Letter · Decision Letter 0]

22 May 2023

PONE-D-23-12558A Bilateral Filtering-based Image Enhancement for Alzheimer Disease Classification using CNNPLOS ONE

Dear Dr. Awarayi,

Thank you for submitting your manuscript to PLOS ONE. After careful consideration, we feel that it has merit but does not fully meet PLOS ONE’s publication criteria as it currently stands. Therefore, we invite you to submit a revised version of the manuscript that addresses the points raised during the review process.

We look forward to receiving your revised manuscript.

Kind regards,

Muhammad Fazal Ijaz

Academic Editor

PLOS ONE

Journal Requirements:

Reviewers' comments:

Reviewer's Responses to Questions

**Comments to the Author**

1. Is the manuscript technically sound, and do the data support the conclusions?

Reviewer #1: Partly

Reviewer #2: Yes

2. Has the statistical analysis been performed appropriately and rigorously? 

Reviewer #1: Yes

Reviewer #2: No

3. Have the authors made all data underlying the findings in their manuscript fully available?

Reviewer #1: No

Reviewer #2: Yes

4. Is the manuscript presented in an intelligible fashion and written in standard English?

Reviewer #1: Yes

Reviewer #2: Yes

5. Review Comments to the Author

Reviewer #1: The overall impression of the technical contribution of the current study is reasonable. However, the Authors may consider making necessary amendments to the manuscript for better comprehensibility of the study.

1. The abstract must be re-written, focusing on the technical aspects of the proposed model, the main experimental results, and the metrics used in the evaluation. Briefly discuss how the proposed model is superior.

2. Additionally, method names should not be capitalized. Moreover, it is not the best practice to employ abbreviations in the abstract, they should be used when the term is introduced for the first time.

3. The contribution of the current study must be briefly discussed as bullet points in the introduction. And motivation must also be discussed in the manuscript.

4. Introduction section in too lengthy, some part of the introduction can be moved as the literature review.

5. Introduction section must discuss the technical gaps associated with the current problem. Authors may refer and include https://doi.org/10.32604/cmc.2021.018472

6. The literature section is missing. Authors are recommended to incorporate the same for better comprehensibility of the study.

7. Authors may provide the architecture/block diagram of the proposed model for better comprehensibility of the proposed model concerning various aspects of the proposed model.

8. What is the size of the input image that is considered for processing and the size of the kernels? Authors may refer and include the content shown in https://doi.org/10.1038/s41598-022-25089-2 for better comprehensibility.

9. Whether authors have used Stride 1 or Stride, 2 must be presented.

10. For how many epochs does the proposed model execute. what is the initial learning rate, and after how many epochs does the model's learning rate saturated.

11. Majority of the figures lack the clarity, they quality is fair but they must be explained in the text and the figures must be cited.

12. By considering the current form of the conclusion section, it is hard to understand by PlosOne Journal readers. It should be extended with new sentences about the necessity and contributions of the study by considering the authors' opinions about the experimental results derived from some other well-known objective evaluation values if it is possible.

13. English proofreading is strongly recommended for a better understanding of the study, and the quality of the figures must be tremendously improved.

14. Captions of the Figures not self-explanatory. The caption of figures should be self-explanatory, and clearly explaining the figure. Extend the description of the mentioned figures to make them self-explanatory. For example : Fig 1. Proposed CNN architecture.

Reviewer #2: 1. Concept addressed in this study is quite good.The contribution of the current study must be briefly discussed as bullet points in the introduction. And motivation must also be discussed in the introduction of the manuscript.

2. The introduction and the literature must be tremendously improvised with a focus on the limitations of existing models.

3.Where is the graph for testing loss and accuracy presented in the study?

4. Please discuss more on the implementation platform and the dataset details as two

sub-sections in the manuscript.

5.Include the block diagram for the proposed approach and its process

6.By considering the current form of the conclusion section, it is hard to understand by the Journal readers. It should be extended with new sentences about the necessity and contributions of the study by considering the authors' opinions about the experimental results derived from some other well-known objective evaluation values if it is possible

6. PLOS authors have the option to publish the peer review history of their article (what does this mean?). If published, this will include your full peer review and any attached files.

Reviewer #1: No

Reviewer #2: **Yes: **Dr.Jana Shafi

---

## [Author Response · Author response to Decision Letter 0]

12 Jul 2023

Editor comments# 3, 4: Data availability

Response: The datasets used in this study were obtained from the Alzheimer’s Disease Neuroimaging Initiative (ADNI) database, a public data repository with access control. Researchers can apply to be granted access to the datasets. Due to the data use policy of the ADNI, the authors do not have absolute control over the data; however, we have duly cited the data source.

The authors would like to state that we do not have control over the dataset due to the data use policy restrictions of the ADNI; however, we will include the link to the database, to which researchers can submit requests to be granted access by the ADNI

Reviewer 1, Comment # 1: The abstract must be re-written, focusing on the technical aspects of the proposed model, the main experimental results, and the metrics used in the evaluation. Briefly discuss how the proposed model is superior.

Response: Thank you for your comments and suggestions. The abstract has been rewritten, taking into account all the suggestions made by the reviewer.

Reviewer 1, Comment # 2: Additionally, method names should not be capitalized. Moreover, it is not the best practice to employ abbreviations in the abstract, they should be used when the term is introduced for the first time

Response: Your comments are duly appreciated: The abbreviations in the abstract have been written in full. Please refer to the abstract section (page 1).

Reviewer 1, Comment # 3,4 and 5: The contribution of the current study must be briefly discussed as bullet points in the introduction. And motivation must also be discussed in the manuscript. Introduction section in too lengthy, some part of the introduction can be moved as the literature review. Introduction section must discuss the technical gaps associated with the current problem.

Response: Thank you for your contribution. The introduction has been modified in accordance to the reviewer’s comments. The literature was imbedded in the introduction making it too lengthy. The literature review in the introduction has, therefore, been moved to a new section (page 2-4). The motivation and technical gaps have been expatiated (page 1). 

Reviewer 1, Comment #6: The literature section is missing. Authors are recommended to incorporate the same for better comprehensibility of the study.

Response: The literature review was added to the introduction and has been moved into a new section for more clarity. See page 2-4. 

Reviewer 1, Comment #7: Authors may provide the architecture/block diagram of the proposed model for better comprehensibility of the proposed model concerning various aspects of the proposed model.

Response: Your suggestions are duly acknowledged. The block diagram has been incorporated for better clarity. 

Reviewer 1, Comment #8: What is the size of the input image that is considered for processing and the size of the kernels? Authors may refer and include the content shown in https://doi.org/10.1038/s41598-022-25089-2 for better comprehensibility.

Response: Thank you for the comments. The size of the input images is defined under the datasets section (page 5) and the convolutional neural network model section (page 7). The kernel filter size is also indicated under the convolutional neural network model section (page 7) and the hyperparameters section (page 8).

Reviewer 1, Comment #9: Whether authors have used Stride 1 or Stride, 2 must be presented.

Response: Stride 1 was used in the convolutional layers and has now been included in the manuscript.

Reviewer 1, Comment #10: For how many epochs does the proposed model execute. what is the initial learning rate, and after how many epochs does the model's learning rate saturated.

Response: Thank you for the comments. The model was trained for 200 epochs with a learning rate of 0.001, which is incorporated under the hyperparameters section. This has been indicated under various sections in the manuscript (Pages 1, 8, and 9)

Reviewer 1, Comment #11: Majority of the figures lack the clarity; they quality is fair but they must be explained in the text and the figures must be cited.

Response: The figure labeling has been revised, and all figures have been cited appropriately for better clarity.

Reviewer 1, Comment #12: By considering the current form of the conclusion section, it is hard to understand by PlosOne Journal readers. It should be extended with new sentences about the necessity and contributions of the study by considering the authors' opinions about the experimental results derived from some other well-known objective evaluation values if it is possible.

Response: Thank you for the comments. The conclusion has been rewritten for better comprehension. (see page 11).

Reviewer 1, Comment #13: English proofreading is strongly recommended for a better understanding of the study, and the quality of the figures must be tremendously improved.

Response: Thank you for the recommendation. Proofreading has been done.

Reviewer 1, Comment #14: Captions of the Figures not self-explanatory. The caption of figures should be self-explanatory, and clearly explaining the figure. Extend the description of the mentioned figures to make them self-explanatory. For example: Fig 1. Proposed CNN architecture.

Response: The figure labeling has been revised, and all figures have been cited appropriately for better clarity.

Reviewer 2, Comment #1,2: Concept addressed in this study is quite good. The contribution of the current study must be briefly discussed as bullet points in the introduction. And motivation must also be discussed in the introduction of the manuscript. The introduction and the literature must be tremendously improvised with a focus on the limitations of existing models

Response: Thank you for your contribution. The introduction and the literature review have been modified in accordance with the reviewer’s comments. The literature review in the introduction has been moved to a new section (pages 2–4) for better clarity. The motivation and technical gaps have been explained (page 1).

comment #3: Where is the graph for testing loss and accuracy presented in the study?

Response: The authors excluded the graphs because the method employed in the training (10-fold cross validation) generated about 10 loss and 10 accuracy graphs, which we thought would be too many to be presented in the manuscript. The training of the model involved 10 iterations, generating a loss and accuracy graph per iteration.

Reviewer 2, Comment #4: Please discuss more on the implementation platform and the dataset details as two sub-sections in the manuscript.

Response: Your comments are duly appreciated. The datasets have been further described under the dataset section, while the implementation platform is presented under the method section.

Reviewer 2, Comment #5: Include the block diagram for the proposed approach and its process.

Response: Thank you for the suggestion. The block diagram has been incorporated for better clarity. 

Reviewer 2, Comment #6: By considering the current form of the conclusion section, it is hard to understand by the Journal readers. It should be extended with new sentences about the necessity and contributions of the study by considering the authors' opinions about the experimental results derived from some other well-known objective evaluation values if it is possible.

Response: Thank you for the comments. The conclusion has been rewritten for better comprehension. (see page 11).

---

## [Decision Letter · Decision Letter 1]

17 Sep 2023

PONE-D-23-12558R1A Bilateral Filtering-based Image Enhancement for Alzheimer Disease Classification using CNNPLOS ONE

Dear Dr. Awarayi,

Thank you for submitting your manuscript to PLOS ONE. After careful consideration, we feel that it has merit but does not fully meet PLOS ONE’s publication criteria as it currently stands. Therefore, we invite you to submit a revised version of the manuscript that addresses the points raised during the review process.

ACADEMIC EDITOR: The reviewers' have reviewed the revised manuscript. Based on the assessment, the paper needs to undergo major revisions as suggested by the reviewers. The authors are requested to make necessary changes to the manuscript and prepare a point-by-point response to the reviewer comments for further consideration. 

We look forward to receiving your revised manuscript.

Kind regards,

Sunder Ali Khowaja, Ph.D.

Academic Editor

PLOS ONE

Reviewers' comments:

Reviewer's Responses to Questions

**Comments to the Author**

1. If the authors have adequately addressed your comments raised in a previous round of review and you feel that this manuscript is now acceptable for publication, you may indicate that here to bypass the “Comments to the Author” section, enter your conflict of interest statement in the “Confidential to Editor” section, and submit your "Accept" recommendation.

Reviewer #3: (No Response)

Reviewer #4: All comments have been addressed

2. Is the manuscript technically sound, and do the data support the conclusions?

Reviewer #3: No

Reviewer #4: Yes

3. Has the statistical analysis been performed appropriately and rigorously? 

Reviewer #3: No

Reviewer #4: N/A

4. Have the authors made all data underlying the findings in their manuscript fully available?

Reviewer #3: Yes

Reviewer #4: Yes

5. Is the manuscript presented in an intelligible fashion and written in standard English?

Reviewer #3: No

Reviewer #4: Yes

6. Review Comments to the Author

Reviewer #3: PONE-D-23-12558R1

I think this article is a revised version but still there many issues that must be considered to properly reach the publication. Some of the suggestions and comments are listed as follows:

1. There are several grammatical mistakes and typos that must be corrected with careful revision, such as from where this parenthesis “….. [1]. ).” Is started, and so on.

2. Most of the abbreviations are incorrect such as Alzheimer’s 8 disease, and so on. Every term should be defined completely for the first time and then use the abbreviation throughout the article, do not repeat.

3. The first sentence “First, an image enhancement technique that effectively reduces noise in MRI 49 images while preserving image quality” of the contribution seems incomplete. Please revise. Additionally, the contribution is very limited.

4. There is no paper organization, please add at the end of the introduction section.

5. Generally, the English writing of this article is very low standard.

6. Moreover, the technical depth of the paper is not adequate too. I strictly recommend the authors to have a look at some standard article and learn about the article writing and organization, etc.

7. Table 1 summarize the existing CNN model used for AD, although there several more latest models such as “3D Convolutional Neural Networks Based Multiclass Classification of Alzheimer’s and Parkinson’s Diseases using PET and SPECT Neuroimaging Modalities, Brain Informatics, 2021”, “On Improved 3D-CNN Based Binary and Multiclass Classification of Alzheimer’s Disease Using Neuroimaging Modalities and Data Augmentation Methods”, Journal of Healthcare Engineering, 2021”, “Early-Stage Alzheimer's Disease Categorization using PET Neuroimaging Modality and Convolutional Neural Networks in the 2D and 3D Domains”, Sensors, 2022”, “On Disharmony in Batch Normalization and Dropout Methods for Early Categorization of Alzheimer's Disease” Sustainability, 2022”. Beside this, the authors can also refer to more latest models for better comparison and understanding.

8. Figure 1 should be redesigned. Most of the arrows are not correctly connected. Also, keep the figures text always consistent with the paper body text.

9. After any equation, in term “Where” w should be small always.

10. Table 4 presents a Comparative analysis of existing and proposed model but the authors should add more latest models in comparison.

11. Most of the reference are old enough and also limited. Further latest references can be added.

Reviewer #4: Authors have addressed the comments made by previous reviewer; however, I would like to suggest that authors should update the literature review with recent studies from 2023, for improved readability and relevancy.

7. PLOS authors have the option to publish the peer review history of their article (what does this mean?). If published, this will include your full peer review and any attached files.

Reviewer #3: No

Reviewer #4: **Yes: **Parus Khuwaja

---

## [Author Response · Author response to Decision Letter 1]

23 Oct 2023

Reviewer 3, Comment # 1: There are several grammatical mistakes and typos that must be corrected with careful revision, such as from where this parenthesis “….. [1]. ).” Is started, and so on.

Response: Thank you for your comments and suggestions. The grammatical mistakes and typos have been rectified. The manuscript has been thoroughly reviewed, and the errors duly corrected. We also used the Grammarly software to check and correct the mistakes.

Reviewer 3, Comment # 2: Most of the abbreviations are incorrect such as Alzheimer’s 8 disease, and so on. Every term should be defined completely for the first time and then use the abbreviation throughout the article, do not repeat.

Response: Your comments are duly appreciated. The corrections have been made to that effect.

Reviewer 3, Comment # 3: The first sentence “First, an image enhancement technique that effectively reduces noise in MRI 49 images while preserving image quality” of the contribution seems incomplete. Please revise. Additionally, the contribution is very limited.

Response: Thanks for the comments. The manuscript has been modified accordingly.

Action: The primary contribution of this study comprises two key components:

• First, an image enhancement technique that effectively reduces noise in MRI images while preserving image quality was designed and implemented. The image enhancement algorithm for color images was developed using two image processing techniques: image equalization and bilateral filtering. The algorithm mainly improves the quality of the images while maintaining their edges and details. The bilateral filtering was used to keep the edges and details of the image, while histogram equalization modifies the intensities of the image to improve its contrast.

• Second, a CNN model consisting of four convolutional layers and two hidden layers successfully classifies AD more accurately than existing deep learning models. The model was trained using the k-fold cross-validation technique, ensuring that all the images were effectively used in training and testing the model.

Reviewer 3, Comment # 4: There is no paper organization, please add at the end of the introduction section.

Response: The paper organization has been added at the end of the introduction.

Action: The remaining sections of the paper are the literature review, materials and methods, results, and discussion. The literature review section presents a review of some existing state-of-the-art studies. A description of the dataset, the proposed method, and a detailed description of the experiment are presented under the materials and methods section. The results section reveals the study's experimental results, which are further discussed in the discussion section.

Reviewer 3, Comment # 5 & 6: Generally, the English writing of this article is very low standard. Moreover, the technical depth of the paper is not adequate too. I strictly recommend the authors to have a look at some standard article and learn about the article writing and organization, etc.

Response: Your comment is duly acknowledged. The manuscript has been reviewed to improve the writing standard. A reference was made to Tufail et al. (2023), Allada et al. (2023) and the manuscript writing guideline of the PLOS ONE journal to gather insight to improve the manuscript.

Reviewer 3, Comment # 7: Table 1 summarize the existing CNN model used for AD, although there several more latest models such as “3D Convolutional Neural Networks Based Multiclass Classification of Alzheimer’s and Parkinson’s Diseases using PET and SPECT Neuroimaging Modalities, Brain Informatics, 2021”, “On Improved 3D-CNN Based Binary and Multiclass Classification of Alzheimer’s Disease Using Neuroimaging Modalities and Data Augmentation Methods”, Journal of Healthcare Engineering, 2021”, “Early-Stage Alzheimer's Disease Categorization using PET Neuroimaging Modality and Convolutional Neural Networks in the 2D and 3D Domains”, Sensors, 2022”, “On Disharmony in Batch Normalization and Dropout Methods for Early Categorization of Alzheimer's Disease” Sustainability, 2022”. Beside this, the authors can also refer to more latest models for better comparison and understanding.

Response: Thanks for your recommendations: We have updated Table 1 with more latest models such as Hridhee et al., (2023), Allada et al., (2023), Gowhar et al., (2023) and Tufail et al., (2022).

Reviewer 3, Comment # 8: Figure 1 should be redesigned. Most of the arrows are not correctly connected. Also, keep the figures text always consistent with the paper body text.

Response: Thanks for the observation and suggestion. Figure 1 has been redesigned to ensure the arrows are well connected, and the text is consistent with the paper’s body text.

Action: 

Reviewer 3, Comment # 9: After any equation, in term “Where” w should be small always.

Response: Thanks for the correction. The correction has been done.

Reviewer 3, Comment # 10: Table 4 presents a Comparative analysis of existing and proposed model but the authors should add more latest models in comparison.

Response: Thank you for the recommendation. Latest models, such as Hridhee et al. (2023), Allada et al. (2023), Gowhar et al. (2023) and Tufail et al. (2022) have been added.

Reviewer 3, Comment # 11: Most of the reference are old enough and also limited. Further latest references can be added.

Response: Thank you for the comment. The suggestion has been implemented.

Action: 

1. Hridhee RA, Bhowmik B, Hossain QD. Alzheimer’s Disease Classification From 2D MRI Brain Scans Using Convolutional Neural Networks. 3rd Int Conf Electr Comput Commun Eng ECCE 2023. 2023;(August):1–6. 

2. Allada A, Bhavani R, Chaduvula K, Priya R. Alzheimer’s disease classification using competitive swarm multi-verse optimizer-based deep neuro-fuzzy network. Concurr Comput Pract Exp. 2023;35(21):1–19. 

3. Mohi ud din dar G, Bhagat A, Ansarullah SI, Othman MT Ben, Hamid Y, Alkahtani HK, et al. A Novel Framework for Classification of Different Alzheimer’s Disease Stages Using CNN Model. Electron. 2023;12(2):1–14. 

4. Tufail A Bin, Anwar N, Othman MT Ben, Ullah I, Khan RA, Ma Y-K, et al. Early-Stage Alzheimer’s Disease Categorization Using PET Neuroimaging Modality and Convolutional Neural Networks in the 2D and 3D Domains. Sensors [Internet]. 2022 Jun 18;22(12):4609. Available from: https://www.mdpi.com/1424-8220/22/12/4609

Reviewer 4, Comment # 11: Authors have addressed the comments made by previous reviewer; however, I would like to suggest that authors should update the literature review with recent studies from 2023, for improved readability and relevancy.

Response: Thank you for your comment and suggestion. Latest models, such as Hridhee et al. (2023), Allada et al. (2023), Gowhar et al. (2023) and Tufail et al. (2022), have been added.

---

## [Decision Letter · Decision Letter 2]

18 Dec 2023

PONE-D-23-12558R2A Bilateral Filtering-based Image Enhancement for Alzheimer Disease Classification using CNNPLOS ONE

Dear Dr. Awarayi,

Thank you for submitting your manuscript to PLOS ONE. After careful consideration, we feel that it has merit but does not fully meet PLOS ONE’s publication criteria as it currently stands. Therefore, we invite you to submit a revised version of the manuscript that addresses the points raised during the review process.

**ACADEMIC EDITOR: **The reviewer reports have been submitted. It seems that the reviewers are suggesting revisions before the manuscript can be considered for publication. The authors are advised to revise the manuscript according to the reviewers comments and prepare a point-by-point response to provide clarity on the revisions been made. 

We look forward to receiving your revised manuscript.

Kind regards,

Sunder Ali Khowaja, Ph.D.

Academic Editor

PLOS ONE

Reviewers' comments:

Reviewer's Responses to Questions

**Comments to the Author**

1. If the authors have adequately addressed your comments raised in a previous round of review and you feel that this manuscript is now acceptable for publication, you may indicate that here to bypass the “Comments to the Author” section, enter your conflict of interest statement in the “Confidential to Editor” section, and submit your "Accept" recommendation.

Reviewer #4: All comments have been addressed

Reviewer #5: All comments have been addressed

Reviewer #6: All comments have been addressed

Reviewer #7: (No Response)

2. Is the manuscript technically sound, and do the data support the conclusions?

Reviewer #4: Yes

Reviewer #5: Yes

Reviewer #6: Partly

Reviewer #7: No

3. Has the statistical analysis been performed appropriately and rigorously? 

Reviewer #4: N/A

Reviewer #5: Yes

Reviewer #6: N/A

Reviewer #7: No

4. Have the authors made all data underlying the findings in their manuscript fully available?

Reviewer #4: No

Reviewer #5: No

Reviewer #6: Yes

Reviewer #7: Yes

5. Is the manuscript presented in an intelligible fashion and written in standard English?

Reviewer #4: Yes

Reviewer #5: No

Reviewer #6: Yes

Reviewer #7: Yes

6. Review Comments to the Author

Reviewer #4: The authors have addressed all the comments in an adequate manner. Therefore, I would like to recommend the acceptance of this article.

Reviewer #5: The author has basically solved my problem, and I have no other problems. Now I recommend receiving the revised manuscript.

Reviewer #6: The article entitled " A Bilateral Filtering-based Image Enhancement for Alzheimer Disease Classification using CNN " can be Accepted in its current form

Reviewer #7: The authors propose Alzheimer Disease (AD) Classification using convolutional neural network. The proposed method employs preprocessing techniques like histogram equalization and bilateral filtering techniques to reduce noise and improve image quality, with the ultimate goal of facilitating classification of AD. Overall contribution of the work is not impressive.

1. The overall representation/structure of paper is not appropriate. It needs figurative approach that should be representation of the overall proposed work.

2. With necessary images show the impact of proposed enhancement techniques (histogram equalization, bilateral filtering).

3. Describe with necessary data how the preprocessing techniques improves the overall accuracy of the proposed work.

4. Compare the result with the standard CNN models like VGG19, ResNet110, Dense net also compare the number of parameters.

5. Data samples are imbalanced how this is handled please describe.

6. Show the confusion matrix for each class and also describe the ROC curves.

7. PLOS authors have the option to publish the peer review history of their article (what does this mean?). If published, this will include your full peer review and any attached files.

Reviewer #4: **Yes: **Parus Khuwaja

Reviewer #5: No

Reviewer #6: No

Reviewer #7: No

---

## [Author Response · Author response to Decision Letter 2]

8 Feb 2024

Reviewer 4, Comment #1: The authors have addressed all the comments in an adequate manner. Therefore, I would like to recommend the acceptance of this article.

Response: Thank you for reviewing and accepting our manuscript.

Reviewer 5, Comment #1: The author has basically solved my problem, and I have no other problems. Now I recommend receiving the revised manuscript.

Response: Thank you for reviewing and accepting our manuscript.

Reviewer 6, Comment #1: The article entitled " A Bilateral Filtering-based Image Enhancement for Alzheimer Disease Classification using CNN " can be Accepted in its current form

Response: Thank you for reviewing and accepting our manuscript.

Reviewer 7, Comment #1: The overall representation/structure of paper is not appropriate. It needs figurative approach that should be representation of the overall proposed work.

Response: Thank you for the comment. We have reviewed the manuscript to ensure it follows the general guidelines of the PLOS ONE journal. We have also introduced more figures to improve the visual elements in the work for better clarity.

Reviewer 7, Comment #2: With necessary images show the impact of proposed enhancement techniques (histogram equalization, bilateral filtering).

Response: Thank you for the suggestion. Images of the proposed enhancement techniques are presented in the Fig 3 of the manuscript. A further comparative assessment of the image quality was performed based on mean pixel intensity, contrast and entropy metrics and presented in Fig 4 of the manuscript.

Reviewer 7, Comment #3: Describe with necessary data how the preprocessing techniques improves the overall accuracy of the proposed work.

Response: Table 2 of the manuscript has been modified to include the test results of the model before the image enhancement. 

Reviewer 7, Comment #4: Compare the result with the standard CNN models like VGG19, ResNet110, Dense net also compare the number of parameters.

Response: Thank you for your recommendation. This comparison has already been done in Table 4 of the manuscript. The study by Hridhee et al., (2023) used VGG16 model while Zhang et al., (2022) used the ResNet110 model.

Reviewer 7, Comment #5: Data samples are imbalanced how this is handled please describe.

Response: Your comments are duly appreciated. Data augmentation was used to increase the data volume; however, in the case of data imbalance from the MCI class, it was observed not to be significant since it did not affect the training process. This can be seen in the confusion matrices in Figs 5 and 6 of the manuscript.

Reviewer 7, Comment #6: Show the confusion matrix for each class and also describe the ROC curves..

Response: Thank for the comment. The confusion matrix and the ROC curves have been included. Refer to the manuscript Figs 5, 6 and 7.

---

## [Decision Letter · Decision Letter 3]

3 Apr 2024

A Bilateral Filtering-based Image Enhancement for Alzheimer Disease Classification using CNN

PONE-D-23-12558R3

Dear Dr. Awarayi,

We’re pleased to inform you that your manuscript has been judged scientifically suitable for publication and will be formally accepted for publication once it meets all outstanding technical requirements.

Kind regards,

Sunder Ali Khowaja, Ph.D.

Academic Editor

PLOS ONE

Additional Editor Comments (optional):

It seems that authors have revised their manuscript in a satisfactory manner as indicated by the reviewer. Therefore, I would like to recommend the acceptance of this manuscript.

Reviewers' comments:

Reviewer's Responses to Questions

**Comments to the Author**

1. If the authors have adequately addressed your comments raised in a previous round of review and you feel that this manuscript is now acceptable for publication, you may indicate that here to bypass the “Comments to the Author” section, enter your conflict of interest statement in the “Confidential to Editor” section, and submit your "Accept" recommendation.

Reviewer #6: All comments have been addressed

2. Is the manuscript technically sound, and do the data support the conclusions?

Reviewer #6: Yes

3. Has the statistical analysis been performed appropriately and rigorously? 

Reviewer #6: N/A

4. Have the authors made all data underlying the findings in their manuscript fully available?

Reviewer #6: Yes

5. Is the manuscript presented in an intelligible fashion and written in standard English?

Reviewer #6: Yes

6. Review Comments to the Author

Reviewer #6: The authors have made substantial corrections in the article . Hence the article can be accepted in its current form.

7. PLOS authors have the option to publish the peer review history of their article (what does this mean?). If published, this will include your full peer review and any attached files.

Reviewer #6: No

---

## [Editor Report · Acceptance letter]

8 Apr 2024

PONE-D-23-12558R3 

PLOS ONE

Dear Dr. Awarayi, 

I'm pleased to inform you that your manuscript has been deemed suitable for publication in PLOS ONE. Congratulations! Your manuscript is now being handed over to our production team.

Kind regards, 

on behalf of

Dr. Sunder Ali Khowaja 

Academic Editor

PLOS ONE